# B Cells with a Senescent-Associated Secretory Phenotype Accumulate in the Adipose Tissue of Individuals with Obesity

**DOI:** 10.3390/ijms22041839

**Published:** 2021-02-12

**Authors:** Daniela Frasca, Maria Romero, Alain Diaz, Denisse Garcia, Seth Thaller, Bonnie B. Blomberg

**Affiliations:** 1Department of Microbiology and Immunology, University of Miami Miller School of Medicine, Miami, FL 33136, USA; mromero5@med.miami.edu (M.R.); a.diaz7@med.miami.edu (A.D.); d.garcia24@umiami.edu (D.G.); bblomber@med.miami.edu (B.B.B.); 2Sylvester Comprehensive Cancer Center, Miami, FL 33136, USA; 3Department of Surgery, Division of Plastic and Reconstructive Surgery, University of Miami Miller School of Medicine, Miami, FL 33136, USA; sthaller@med.miami.edu

**Keywords:** B cells, senescence, inflammation, obesity

## Abstract

Senescent cells accumulate in the adipose tissue (AT) of individuals with obesity and secrete multiple factors that constitute the senescence-associated secretory phenotype (SASP). This paper aimed at the identification of B cells with a SASP phenotype in the AT, as compared to the peripheral blood, of individuals with obesity. Our results show increased expression of SASP markers in AT versus blood B cells, a phenotype associated with a hyper-metabolic profile necessary to support the increased immune activation of AT-derived B cells as compared to blood-derived B cells. This hyper-metabolic profile is needed for the secretion of the pro-inflammatory mediators (cytokines, chemokines, micro-RNAs) that fuel local and systemic inflammation.

## 1. Introduction

Cellular senescence indicates the irreversible arrest of cell proliferation that is induced by different stress-derived signals. It is mediated by the inhibition of cell cycle progression through p16^INK4^ and/or the activation of cell cycle arrest through p53/p21. Stressors inducing cell senescence include DNA damage, telomere shortening, radiation, reactive metabolites, mitogenic and metabolic stressors. Although senescent cells show changes in chromatin organization and gene expression, they remain metabolically active [1,2]. The senescence-associated secretory phenotype (SASP) of senescent cells is characterized by the secretion of soluble pro-inflammatory factors (cytokines, chemokines, micro-RNAs), soluble cytokine receptors (TNF receptors), non-protein soluble factors (nitric oxide), growth factors (EGF, VGEF, NGF) and extracellular matrix macromolecules (fibronectin, collagens, laminin) [3,4]. Senescent cells accumulate in the body during aging, promote tissue degeneration and malignant transformation, and lead to the development of inflammatory-based age-associated diseases [5,6]. Senescent cells also accumulate in the adipose tissue (AT) of mice and humans, and secrete multiple SASP factors that have been shown to induce cell death, increased local and systemic inflammation and recruitment of immune cells, leading to AT dysfunction, insulin resistance and type-2 diabetes [7,8,9].

Obesity is a condition associated with chronic low-grade systemic inflammation, known as inflammaging [10]. Inflammaging induces chronic immune activation (IA), functional impairment of immune cells and decreased immunity. Obesity and associated inflammation lead to several debilitating chronic diseases such as type-2 diabetes [11,12,13], cardiovascular disease [14], cancer [15], atherosclerosis [16], inflammatory bowel disease [17]. Obesity also leads to dysfunctional immunity and decreases the proportion of individuals responding to vaccination [18,19,20], as well as of patients achieving remission in response to therapy [21,22,23], which may further affect the outcome of infection, with obvious health consequences for the individual. It is not currently known if the obesity-driven impairment of immune function contributes to reduced clearance of senescent cells leading to their increased accumulation in the AT.

We have previously characterized the obesity-associated decrease in B cell function [18]. We have shown that unstimulated B cells from the peripheral blood of individuals with obesity, as compared to those from lean individuals, express higher levels of RNA and protein for multiple inflammaging-associated pro-inflammatory cytokines (e.g., TNF-α). B cell intrinsic TNF-α positively correlates with serum TNF-α and both negatively correlate with B cell function, measured by activation-induced cytidine deaminase (AID) after in vivo or in vitro stimulation with mitogens, antigens and vaccines. AID is the enzyme that regulates Ig (immunoglobulin) class switch recombination and somatic hypermutation [24], two processes leading to the generation of high affinity protective antibodies [25,26,27].

The accumulation of senescent T cells in the human AT under obesity conditions has been reported [28], but virtually nothing is known about the accumulation of senescent B cells. In this paper, we have investigated senescent B cells present in the subcutaneous AT of patients undergoing weight reduction surgeries, as compared to those from the blood of individuals with obesity (age- and gender-matched), and we have characterized their SASP phenotype and their metabolic profile. Results show increased expression of SASP markers in AT versus blood B cells, a phenotype associated with a hyper-metabolic profile necessary to support increased IA of B cells from the AT as compared to those from the peripheral blood, including the production of several pro-inflammatory mediators that fuel local and systemic inflammation.

## 2. Results

### 2.1. Increased Expression of SASP Markers in B Cells from the AT of Individuals with Obesity as Compared to the Peripheral Blood

B cells were isolated from the peripheral blood of individuals with obesity using magnetic beads and from the AT of obese surgery patients using flow cytometry and cell sorting (as indicated in Materials and Methods, Section 4.4). After isolation, B cells were left unstimulated and were resuspended in TRIzol. The RNA was then extracted and the expression of SASP markers evaluated by qPCR. We measured RNA expression of the following pro-inflammatory markers: cytokines (TNF, IL-6), chemokines (IL-8), micro-RNAs, miRs (miR-155, miR-16, miR-181a) and cell cycle inhibitors and markers of proliferation arrest (p16^INK4^, p21^CIP1/WAF1^ and p53). Figure 1 shows significantly higher expression of RNA for pro-inflammatory cytokines and chemokines (1A), for pro-inflammatory miRs (1B) and for p16^INK4^, p21^CIP1/WAF1^ and p53 (1C) in B cells from the AT as compared to B cells from the peripheral blood.

Although the purpose of our study was to compare RNA levels of multiple SASP markers in B cells from the AT of surgery patients and from the blood of individuals with obesity, we also evaluated RNA expression of these SASP markers in B cells from the blood of lean individuals as compared to our cohort of obese individuals (age- and gender-matched). We found significantly increased RNA expression of all SASP markers in B cells from obese versus lean individuals: TNF, 0.38 ± 0.03 versus 0.17 ± 0.03 (*p* = 0.0006); IL-6, 0.23 ± 0.01 versus 0.16 ± 0.01 (*p* = 0.003); IL-8, 0.17 ± 0.01 versus 0.11 ± 0.01 (*p* = 0.0008); miR-155, 0.23 ± 0.03 versus 0.09 ± 0.02 (*p* = 0.001); miR-16, 0.31 ± 0.03 versus 0.11 ± 0.02 (*p* = 0.0009); miR-181a, 0.18 ± 0.01 versus 0.08 ± 0.01 (*p* = 0.0009); p16^INK4^, 0.28 ± 0.02 versus 0.15 ± 0.02 (*p* = 0.001); p21^CIP1/WAF1^, 0.30 ± 0.01 versus 0.19 ± 0.02 (*p* = 0.0001); p53, 0.41 ± 0.04 versus 0.15 ± 0.01 (*p* = 0.0002). These results altogether clearly indicate that the blood of individuals with obesity, as compared to the blood of lean individuals, is enriched in B cells expressing RNA for multiple SASP markers. These markers are expressed at even higher levels in B cells from the AT of surgery patients.

### 2.2. Increased Frequencies of Memory B Cells, and Decreased Frequencies of Naïve B Cells, in the AT of Individuals with Obesity as Compared to the Peripheral Blood

Our previously published findings on the expression of SASP markers in B cells from the peripheral blood of young and elderly individuals have shown that these markers are almost exclusively expressed in memory B cells. We therefore measured the frequencies of naïve and memory B cell subsets in the blood and in the AT of our participants. We used anti-IgD and anti-CD27 antibodies to identify the four major subsets of B cells: naïve (IgD + CD27−), IgM memory (IgD + CD27+), switched memory (IgD-CD27+), and double negative memory (DN, IgD-CD27−) B cells. Results in Figure 2A show gating strategies to evaluate frequencies of naïve and memory B cells in the blood and in the AT of individuals with obesity. Results in Figure 2B show higher frequencies of memory B cells [that include the subsets of IgM memory, switched memory and DN B cells], and reduced frequencies of naïve B cells, in the AT as compared to the blood of individuals with obesity. Naïve B cell frequencies in the blood of obese versus lean individuals were decreased, although not significantly (42 ± 4 versus 53 ± 4, *p* = 0.05), whereas memory B cells were significantly higher (58 ± 2 versus 47 ± 3 (*p* = 0.04). These results altogether suggest that the increased expression of SASP markers in B cells from the blood of obese versus lean individuals, as well as in B cells from AT versus blood B cells, is likely dependent on the increased frequency of memory B cell subsets.

### 2.3. B Cells from the AT of Individuals with Obesity Are Highly Metabolic as Compared to Those from the Peripheral Blood

Senescent cells have been shown to be metabolically active, a condition necessary to support the SASP. We evaluated the metabolic profile of B cells from the blood and the AT of individuals with obesity, which depends on uptake of nutrients. We, herefore, measured the uptake of glucose and lipids, by flow cytometry, using the glucose analog (2-(N-(7-Nitrobenz-2-oxa-1,3-diazol-4-yl)Amino)-2-Deoxyglucose) (2-NBDG) and the Deep Red Neutral Lipid compound LipidTOX, respectively. Results show increased glucose uptake (Figure 3A) and increased lipid uptake (Figure 3B) in B cells from the AT as compared to B cells from the blood. When we compared glucose uptake in B cells from the blood of lean and obese individuals, we found significantly increased 2-NBDG staining in B cells from obese versus lean individuals (1073 ± 46, versus 741 ± 40, *p* = 0.0003). Similar results were obtained when we measured lipid uptake by LipidTOX staining (8071 ± 353 versus 5804 ± 615, *p* = 0.004). Results in Figure 3C also show increased RNA expression of enzymes involved in metabolic pathways, such as HK2, hexokinase 2, a key glycolytic enzyme that phosphorylates glucose; LDHA, lactate dehydrogenase, that converts pyruvate into lactate and represents a measure of anaerobic glycolysis; PDHX, a component of the pyruvate dehydrogenase complex that converts pyruvate into acetyl-CoA and represents a measure of oxidative phosphorylation and mitochondrial function; ACACB, acethyl-CoA carboxylase, a regulator of fatty acid synthesis. Values in B cells from the blood of obese versus lean individuals were: HK2, 0.27 ± 0.01 versus 0.18 ± 0.01 (*p* = 0.0001); LDHA, 0.18 ± 0.01 versus 0.10 ± 0.001 (*p* = 0.0001); PDHX, 0.18 ± 0.012 (*p* = 0.0001) versus 0.12 ± 0.01; ACACB, 0.29 ± 0.01 versus 0.25 ± 0.01 (*p* = 0.005). These results altogether suggest that obesity induces B cells characterized by a higher metabolic profile which is needed to support their increased IA and function, including the production of pro-inflammatory mediators that support local and systemic inflammation. The higher metabolic phenotype of B cells from the AT is also associated with increased secretion of IgG antibodies with autoimmune specificity, as we have previously shown [29,30].

## 3. Discussion and Conclusions

The comprehensive investigation of the contribution of senescent B cells to AT dysfunction is warranted. Results in this manuscript have identified and characterized B cells with a SASP phenotype in the AT of patients undergoing weight reduction surgeries, as compared to those from the peripheral blood of individuals with obesity (age- and gender-matched). Results show increased expression of several SASP markers (pro-inflammatory cytokines, chemokines and miRs, as well as markers of cell cycle arrest) in B cells from the blood of obese versus lean individuals, confirming and extending our previously published findings [18]. Moreover, these SASP markers are expressed at higher levels in B cells from the AT as compared to those from the peripheral blood, providing the first evidence of the presence of senescent B cells in the human AT. Our results also show that AT-derived B cells are hyper-metabolic as compared to blood-derived B cells. This metabolic reprogramming is necessary to support increased immune activation and secretion of SASP factors that fuel local and systemic inflammation, as it has already been shown [31]. Cells with a senescent phenotype are known to have an altered metabolism associated with increased oxidative stress, impairment of specific metabolic pathways and accumulation of oxidized proteins [32]. Metabolic reprogramming is therefore required for these cells to cope with the energetic demands of the senescent program that include the increased secretion of SASP factors, increased oxidative stress and increased endoplasmic reticulum stress.

Cellular senescence indicates the cell cycle arrest elicited in response to a variety of stressors. Despite their inability to proliferate, senescent cells are transcriptionally and metabolically active and secrete multiple SASP factors [3]. In this paper we measured TNF and IL-6 RNA expression because these pro-inflammatory cytokines are secreted at high levels by both the adipocytes and the immune cells in the obese AT, and regulate lipolysis and local inflammation, as we have previously demonstrated [30]. We measured RNA expression of the chemokine IL-8 because is secreted in large amounts by the obese AT [30] and is responsible for the recruitment of both naïve and memory B cell subsets [33]. For the miRs, we selected miR-155, miR-16 and miR-181a because they regulate AT inflammation and adipocyte differentiation [34,35,36]. We also measured RNA expression of the canonical marker of senescence p16^INK4^, associated with cell cycle arrest in senescent B cells, as we have previously shown [37], as well as of the other cell cycle regulators p21^CIP1/WAF1^ and p53.

The AT of obese human beings is characterized by increased oxidative stress, measured by accumulation of intracellular reactive oxygen species (ROS) [38], known to accelerate senescence of AT-derived cells, as indicated by the finding that intracellular ROS activates p38 mitogen-activated protein kinases (MAPKs), which in turn induce p53/p21^CIP1/WAF1^-dependent senescence [39]. Telomere shortening, another mechanism of cell senescence, has also been shown in the AT [40]. Although the majority of studies have evaluated the senescent phenotype of non-immune cells (pre-adipocytes and adipocytes) in the obese AT, we show here that AT-derived B cells also have a senescent phenotype, confirming recent findings showing high inflammatory mouse AT-B cells, whose expansion is dependent on the activation of the NLRP3 inflammasome [41].

B cells are among the first immune cell types that infiltrate the obese AT in both mice [41,42,43] and humans [30], where they secrete pro-inflammatory mediators, regulate inflammatory T cells and macrophages, and secrete pathogenic IgG antibodies with autoimmune specificities [29,43]. Therefore, depletion of systemic [43] or AT-derived [41] B cells, through intra-peritoneal or intra-AT injection with an anti-CD20 depleting monoclonal antibody, has shown to significantly improve metabolic and immunological function of the tissue in mice. In both cases, mice injected with the anti-CD20 antibody showed a reduction in pathogenic B cell numbers and function, as well as increased glucose and insulin tolerance. Moreover, consistent with a role for B cells in regulating the AT microenvironment through the modulation of inflammatory T cells and macrophages, significantly decreased secretion of the key pro-inflammatory mediators IFN-γ and TNF-α was observed in the AT of anti-CD20-treated mice as compared to isotype-treated controls. These results altogether demonstrate the importance of B cell depletion therapies and support the need to identify additional ways to block hyper-inflammatory and hyper-metabolic B cells that feed-forward local and systemic inflammation, and alleviate obesity-associated metabolic and immunological dysfunction. This represents an important step towards the improvement of the biological quality of life in the obese population.

In conclusion, our results have identified and characterized senescent B cells in the AT of individuals undergoing weight reduction surgery, as compared to those in the peripheral blood of individuals with obesity, thus providing the first evidence of the presence of senescent B cells in the human obese AT. Our results have also identified dysregulated metabolic pathways associated with the SASP. These pathways/markers may be targeted to improve immune function not only in the obese population but also in the general population. Importantly, these interventions will have a significant impact in the management of obesity and related comorbidities, reducing the risk of chronic disease and adverse health outcomes in this vulnerable population.

## 4. Materials and Methods

### 4.1. Subjects

Experiments were conducted using the obese subcutaneous AT obtained from adult female donors undergoing breast reduction surgery [*n* = 8, age = 40–55 years, body-mass index (BMI) ≥ 35 kg/m^2^] at the Division of Plastic and Reconstructive Surgery at the University of Miami Hospital. As controls, we used peripheral blood from age-, gender- and BMI-matched individuals with obesity (*n* = 18). Study participants provided written informed consent. The study was reviewed and approved by the Institutional Review Board (IRB, protocols #20070481 and #20160542), which reviews all human research conducted under the auspices of the University of Miami. We enrolled participants without cancer, Congestive Heart Failure, Cardiovascular Disease, Chronic Renal Failure, renal or hepatic diseases, autoimmune diseases, infectious disease as well as individuals without recent (<3 months) trauma or surgery, pregnancy, or documented current substance abuse.

### 4.2. Isolation and Processing of the AT

The AT isolated from surgery patients was harvested, weighed and washed with 1× Hanks’ Balanced Salt Solution (HBSS). It was then resuspended in Dulbecco’s modified Eagle’s medium (DMEM), minced into small pieces, passed through a 70 μm filter and digested with collagenase type I (SIGMA C-9263) for 1 h in a 37 °C water bath. Digested cells were passed through a 300 μm filter, centrifuged at 300× *g* in order to separate the floating adipocytes from the stromal vascular fraction (SVF) containing the immune cells. The cells floating on the top were transferred to a new tube as adipocytes. The cell pellet (SVF) on the bottom was resuspended in ACK for 3 min at RT (room temperature) to lyse the Red Blood Cells. The SVF was washed 3 times with DMEM. Nucleated cells were counted in a hemocytometer.

### 4.3. Flow Cytometry

Peripheral Blood Mononuclear Cells (PBMC) and SVF from obese donors (10^6^ cells) were stained for 20 min at room temperature (RT) with the following antibodies: Live/Dead detection kit (InVitrogen1878898), anti-CD45 (Biolegend 368540) and anti-CD19 (BD 555415). To evaluate naïve and memory B cells, PBMC and SVF were stained with anti-CD19, anti-CD27 (BD 555441) and anti-IgD (BD 555778) to measure naive (IgD + CD27-) and all memory B cells that include the subsets of IgM memory (IgD + CD27+), switched memory (IgD-CD27+), and double negative memory (DN, IgD-CD27-) B cells. After staining, red blood cells were lyzed using the RBC Lysing Solution (BD 555899), according to the manufacturer’s instructions. Up to 10^5^ events in the lymphocyte gate were acquired on an LSR-Fortessa (BD) and analyzed using FlowJo 10.5.3 software. Single color controls were included in every experiment for compensation. Isotype controls were also used in every experiment to set up the gates.

### 4.4. B Cell Sorting

PBMC from individuals with obesity were collected using Vacutainer CPT tubes (BD 362761) and cryopreserved. PBMC (1 × 10^6^/mL) were thawed and cultured in complete medium (c-RPMI, RPMI 1640, supplemented with 10% FCS, 10 µg/mL Pen-Strep, 1mM Sodium Pyruvate, and 2 × 10^–5^ M 2-ME and 2 mM L-glutamine). B cells were isolated from PBMC using magnetic CD19 Microbeads (Miltenyi), following manufacturer’s instructions.

For B cell isolation from the AT, the SVF was stained with anti-CD45 and anti-CD19 antibodies. CD19+ B cells were sorted in a Sony SH800 cell sorter. Cell preparations were typically >98% pure.

### 4.5. RNA Extraction and Quantitative (q)PCR

To evaluate the expression of SASP markers, B cells were resuspended in TRIzol (Ambion) (10^6^ cells/100 µL), then RNA extracted for quantitative (q)PCR. Total RNA was isolated according to the manufacturer’s protocol, eluted into 10 µL of pre-heated elution buffer and stored at −80 °C until use. Reverse transcriptase (RT) reactions were performed in a Mastercycler Eppendorf Thermocycler to obtain cDNA. Briefly, 10 µL of mRNA or 2 µL of RNA at the concentration of 0.5 µg/µL were used as template for cDNA synthesis in the RT reaction. For miRs quantification, RNA was reverse transcribed in the presence of specific primers (provided together with the qPCR primers, see below). In both cases, conditions were: 40 min at 42 °C and 5 min at 65 °C.

To evaluate RNA expression of enzymes involved in metabolic pathways, the mRNA was extracted from B cells, using the µMACS mRNA isolation kit (Miltenyi), according to the manufacturer’s protocol, eluted into 75 µL of pre-heated elution buffer, and stored at −80 °C until use. The mRNA was reverse transcribed with the same conditions as above.

qPCR reactions were conducted in MicroAmp 96-well plates and run in the ABI 7300 machine. Calculations were made with ABI software. Briefly, we determined the cycle number at which transcripts reached a significant threshold (Ct) for each target gene, and for GAPDH or U6 as controls. The difference in Cts between the housekeeping genes (GAPDH or U6) and the target genes was calculated as ΔCt. Then the relative amount of the target gene was expressed as 2^−ΔCt^ and indicated as qPCR values. Reagents and Taqman primers, all from Life Technologies, were the following: GAPDH, Hs99999905_m1; TNF, Hs01113624_g1; IL-6, Hs00985639_m1; IL-8, Hs00174103_m1; p16^INK4^ (CDKN2A), Hs00923894_m1; p21^CIP1/WAF1^, Hs00355782_m1; p53, Hs01034249_m1. HK2, Hs00606086_m1; LDHA, Hs01378790_g1; PDHX, Hs00185790_m1; ACACB, Hs01565914_m1; U6, 001973; miR-155, 002623; miR-16, 000391; miR-181a, 000480.

### 4.6. Glucose and Lipid Uptake Measurements

PBMCs (10^6^/mL) and SVF were stained with the fluorescent glucose analog (2-(N-(7-Nitrobenz-2-oxa-1,3-diazol-4-yl)Amino)-2-Deoxyglucose) (2-NBDG, Thermo Fisher N13195), or with the Deep Red Neutral Lipid Stain LipidTOX (Thermo Fisher H34476), for 30 min at RT, at the final concentrations recommended by the manufacturer. Cells were then washed and stained for 20 min at room temperature with anti-CD45, anti-CD19, as well as with the Live/Dead detection kit. Cells were washed and later acquired in a BD LSR Fortessa Flow cytometry instrument, using the FITC channel to detect the signal from the fluorescent glucose uptake tracker and the APC channel to detect the signal from the LipidTOX. Fluorescence data were analyzed using FlowJo 10.0.6 software.

### 4.7. Statistical Analyses

Mean comparisons were performed by unpaired Student’s *t* tests (two-tailed), using GraphPad Prism version 8 software, which was used to construct all graphs.

## Figures and Tables

**Figure 1 ijms-22-01839-f001:**
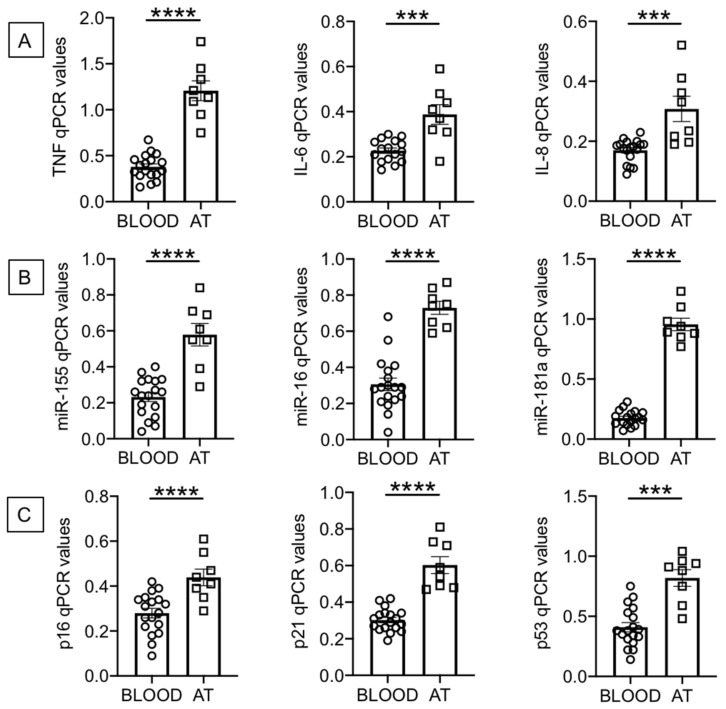
Increased expression of senescence-associated secretory phenotype (SASP) markers in B cells from the adipose tissue (AT), as compared to those from the peripheral blood, of individuals with obesity. B cells isolated from peripheral blood mononuclear cells (PBMC) and stromal vascular fraction (SVF) were resuspended in TRIzol, then the RNA was extracted and the expression of SASP markers detected by qPCR to evaluate expression of RNA for pro-inflammatory cytokines and chemokines (**A**), pro-inflammatory miRs (**B**), and cell cycle regulators p16^INK4^, p21^CIP1/WAF1^ and p53 (**C**). qPCR values are measures of RNA expression of target genes, relative to the housekeeping genes GAPDH or U6 (for miRs quantification), calculated as 2^−ΔCts^. Mean comparisons between groups were performed by unpaired Student’s *t* test (two-tailed). *** *p* < 0.001, **** *p* < 0.0001.

**Figure 2 ijms-22-01839-f002:**
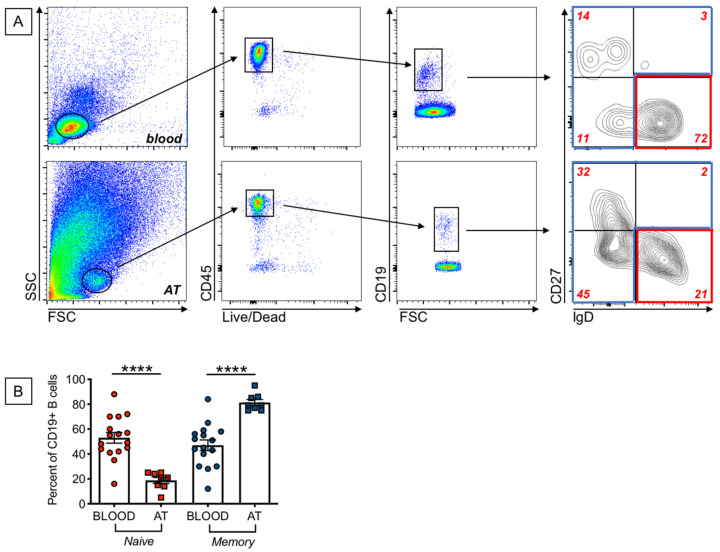
Increased frequencies of memory B cells, and decreased frequencies of naïve B cells, in the AT, as compared to the peripheral blood, of individuals with obesity. (**A**). Gating strategies and a representative dot plot from PBMC (**top**) and SVF (**bottom**) to show naïve and memory B cell subsets. (**B**). Frequencies of naïve and the memory B cell subsets that include IgM memory, switched memory and DN memory B cells. Mean comparisons between groups were performed by Student’s *t* test (two-tailed). **** *p* < 0.0001.

**Figure 3 ijms-22-01839-f003:**
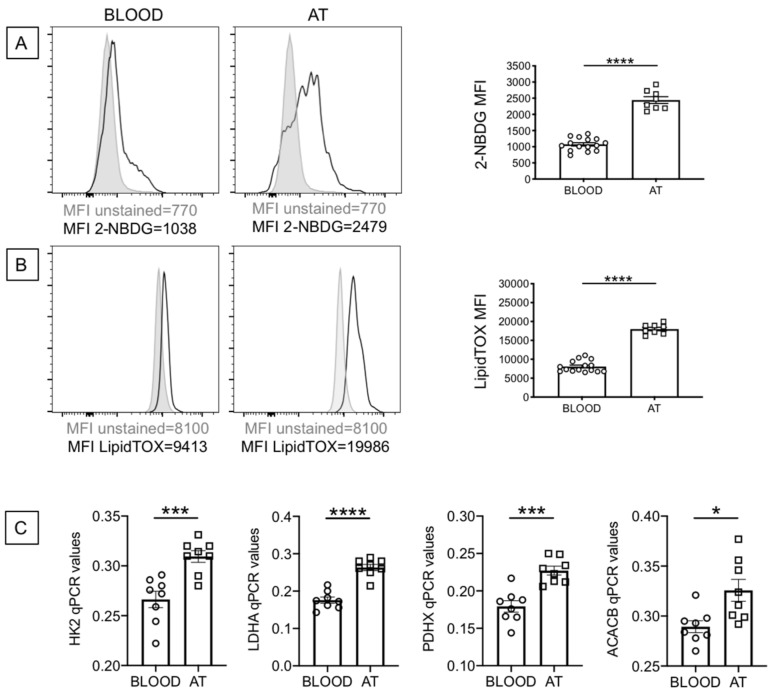
B cells from the AT of individuals with obesity are highly metabolic as compared to those from the peripheral blood. (**A**) Glucose uptake was measured by flow cytometry and the glucose fluorescent analog 2-NBDG.Results show the MFI (mean fluorescence intensity) profile of one representative blood donor and one representative AT donor (**left 2 panels**) and data from all donors (**right**). (**B**) Lipid uptake was measured by flow cytometry and the Deep Red Neutral Lipid Stain LipidTOX. Results show the MFI profile of one representative blood donor and one representative AT donor (**left 2 panels**) and data from all donors (**right**). (**C**) The mRNA was extracted and qPCR performed to evaluate expression of HK2, LDHA, PDHX and ACACB. qPCR values are measures of RNA expression of target genes, relative to the housekeeping gene GAPDH, calculated as 2^−ΔCts^. Mean comparisons between groups were performed by Student’s *t* test (two-tailed). * *p* < 0.05, *** *p* < 0.001, **** *p* < 0.0001.

## Data Availability

The data presented in this study are available upon request to the corresponding author.

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
