# Peer review of "B Cells with a Senescent-Associated Secretory Phenotype Accumulate in the Adipose Tissue of Individuals with Obesity"

_ijms, 2021, doi:10.3390/ijms22041839_

Round 1

Reviewer 1 Report

After the revision the quality of the manuscript has been improved and all my concerns have been addressed. I therefore recommend the publication of the paper in the current form. 

Reviewer 2 Report

The authors have addressed my comments and the paper can be accepted.

This manuscript is a resubmission of an earlier submission. The following is a list of the peer review reports and author responses from that submission.

Round 1

Reviewer 1 Report

In this paper, Daniela Frasca and co-workers showed very interesting data about senescent B cells in the adipose tissue (AT). Very interestingly, markers related to inflammation and senescence increased in B cells from the AT, in comparison to the PB. This aspect is most likely due to the accumulation of memory/terminally differentiated B cells in the AT, which supports inflammation and senescence. This study is in line with the previous work of the lab and gives another clear indication that cellular senescence may exist in B cells in the AT. Furthermore, the accumulation of senescent B cells may contribute to age-related inflammation (inflammaging). For this reason, I believe that these data must certainly be published as they are of great interest.

I have just few minor suggestions to improve the manuscript:

  • TNF (and not TNF-α) is the right term
  • The Figure reporting the gating strategy should be put before the one describing the expression of pro-inflammatory markers. Indeed already at this stage should be clear to readers how the authors defined B cells
  • Line 83: it must be reported how memory and naïve B cells are defined, as it is the first time that authors mentioned them
  • Figure 2A: in the figure reporting the gating strategy for AT the plots CD27/IgD is completely white.

Reviewer 2 Report

12/29/2020  

Major points

  • The aims of the study are missing and are not clear. The authors explain that they previously compared the function of B cells in obese and healthy individuals. However, they should elaborate more on the main aims of this paper in the introduction (lines 54-59). They should connect better the concepts of senescence, immunity and inflammation with the specific aims of this paper.

  • If the primary aim of the study was to investigate senescence (and not only the pro-inflammatory phenotype of B cells in obesity), it would be interesting to analyze also other markers of senescence other than p16 mRNA. For example, the authors could analyze the protein level of p16, p21, p53. Moreover, the authors should provide evidence that the cytokines and chemokines are actually secreted or translated into proteins, instead of showing only the increase in RNA expression.

  • The authors compared B cells from adipose tissue and blood of obese individuals, however the healthy controls are missing. There are no healthy individuals so the outcome of this study is limited. The authors mention that they previously studied the differences in function and activation of B cells between obese and healthy individuals. What about the differences in B cell metabolic activation between healthy and obese individuals (paragraph 2.3)? Did they check it before? Still, they didn’t include these data in the same paper.

  • Samples obtained from blood and adipose tissues do not derive from the same individuals but from different groups (materials and methods).

  • The authors suggest that the SASP factors produced by B cells in the adipose tissue are important to maintain the chronic inflammation associated with obesity. However, the authors focus only on the impact of SASP on the activation and metabolism of B cells themselves, and they did not analyze, for example, the correlation of SASP with the metabolism and senescence of local adipocytes.

  • Figure 3C: the increase in RNA expression seems to be very modest (even though the unit used on Y axis is ambiguous so it is not clear). I don’t think these data are sufficient to suggest an increase in the rate of these metabolic pathways.

  • Discussion: the discussion focuses only on the previous literature. There is no discussion of the results, future directions or physiological relevance.

Minor points:

  • Lines 120 to 123 must be deleted.
  • what’s the unit on Y axis on qPCR charts? I don’t get what “qPCR values” means
  • References have double index numbers